

# Measuring inter-individual differences in behavioural types of gilthead seabreams in the laboratory using deep learning

Marco Signaroli, Arancha Lana, Martina Martorell-Barceló, Javier Sanllehi, Margarida Barcelo-Serra, Eneko Aspillaga, Júlia Mulet and Josep Alós

Fish Ecology Group, Instituto Mediterráneo de Estudios Avanzados, IMEDEA (CSIC-UIB), Esporles, Illes Balears, Spain

## ABSTRACT

Deep learning allows us to automatize the acquisition of large amounts of behavioural animal data with applications for fisheries and aquaculture. In this work, we have trained an image-based deep learning algorithm, the Faster R-CNN (Faster region-based convolutional neural network), to automatically detect and track the gilthead seabream, *Sparus aurata*, to search for individual differences in behaviour. We collected videos using a novel Raspberry Pi high throughput recording system attached to individual experimental behavioural arenas. From the continuous recording during behavioural assays, we acquired and labelled a total of 14,000 images and used them, along with data augmentation techniques, to train the network. Then, we evaluated the performance of our network at different training levels, increasing the number of images and applying data augmentation. For every validation step, we processed more than 52,000 images, with and without the presence of the gilthead seabream, in normal and altered (*i.e.*, after the introduction of a non-familiar object to test for explorative behaviour) behavioural arenas. The final and best version of the neural network, trained with all the images and with data augmentation, reached an accuracy of 92,79% ± 6.78% [89.24–96.34] of correct classification and 10.25 ± 61.59 pixels [6.59-13.91] of fish positioning error. Our recording system based on a Raspberry Pi and a trained convolutional neural network provides a valuable non-invasive tool to automatically track fish movements in experimental arenas and, using the trajectories obtained during behavioural tests, to assay behavioural types.

Corresponding author
Marco Signaroli,
msignaroli@imedea.uib-csic.es

## INTRODUCTION

Deep learning (DL) resulted in a real revolution in many scientific disciplines (*Goh, Hodas & Vishnu, 2017*; *Heaton, Polson & Witte, 2017*; *Lample & Chaplot, 2017*; *Navares & Aznarte, 2020*; *Shen, Wu & Suk, 2017*) and it appeared suddenly in all fields of ecology to address a great variety of challenges (*Christin, Hervet & Lecomte, 2019*). For example, ecologists have been using DL for species identification and classification (*Villon et al., 2018*), density and diversity estimation (*Lee, Chung & Hwang, 2016*; *Salman et al., 2019*), environmental conservation (*Burguera, 2020*; *Lamba et al., 2019*),

resource management (*Hartill et al., 2020*) and animal body size estimations (*Álvarez Ellacuría et al., 2019*). For some fields in ecology, like behavioural ecology, DL represented a real turning point. In fact, behavioural ecology has historically been a mainly qualitative science, because of the difficulty of measuring behavioural parameters, which imply many degrees of freedom (*Brown & De Bivort, 2018*). In the last decades, however, the fast development of technology and computational power allowed us to handle high-dimensional variables, gradually enhancing our capacity to quantify behavioural traits (*Christin, Hervet & Lecomte, 2019*; *Dell et al., 2014*). In recent times, the major bottleneck consists in managing this large and complex amount of data. The analyses are time-consuming and almost always subject to discrepancies between the measurements of different observers (*Webb, 2018*). Using DL, we can overcome these challenges by automatically measuring and quantifying animals' behaviour with an unprecedented level of detail. DL algorithms have been used in behavioural ecology, for example, to predict and classify worms' behaviour (*Li et al., 2017*), to simulate courtship rituals in monogamous species (*Wachtmeister & Enquist, 2000*) and to study the evolution of species' recognition in sympatry (*Ryan & Getz, 2000*).

The main revolution in behavioural ecology has been fostered by the emergence of convolutional neural networks (CNNs), a class of deep learning algorithms developed specifically for object detection and classification in images and videos (*Goodfellow, Bengio & Courville, 2016*). Image acquisition and video recording have been widely used in behavioural ecology, because of their efficiency in capturing relevant information such as body postures or individual movements. There are countless types of video recording applications, from motion-sensor camera traps placed in the wild to take populations censuses (*Norouzzadeh et al., 2018*), to tracking experiments in artificial behavioural arenas to study collective movements (*Romero-Ferrero et al., 2019*). However, the traditional computer vision algorithms used to detect objects in images and videos require manual and problem-specific feature selection and extraction (colours, edges, corners, etc.), limiting our experimental possibilities and making laborious to readapt the algorithm when conditions change (*Zhou et al., 2019*; *Niu et al., 2018*). In the laboratory, this could prevent us from altering the behavioural arenas, for example, by introducing novel objects or by adding enrichment elements, that are highly recommended in ecological studies of behaviour (*Bengston, Pruitt & Riechert, 2014*; *Roberts, Taylor & Garcia De Leaniz, 2011*; *Würbel, 2001*; *Jones, Webster & Salvanes, 2021*).

In this context, CNNs have emerged as an optimal solution: instead of filtering images for the target objects using manual-selected feature information; CNNs are goal-oriented algorithms that automatically learn and extract relevant feature patterns from raw data (labelled images) during a previous training phase (*Goodfellow, Bengio & Courville, 2016*). This new family of algorithms have outmatched all the previous attempts at detecting and classifying objects in images, allowing to track animals in sub-optimal light conditions and in complex and moving environments. CNNs have been used, for example, to describe and catalogue the activity of wild animals from images obtained by camera traps (*Gomez Villa, Salazar & Vargas, 2017*; *Tabak et al., 2019*), to automatically detect and quantify different species of jellyfish (*Pelagia nocticula*, *Cotylorhiza tuberculata* and *Rhizostoma pulmo*) from

videos (*Martin-Abadal et al., 2020*), to focus on the motion of specific body parts of mice (*Mus musculus*) and *Drosophila* (*Mathis et al., 2018*) and to track individual and group movements of zebrafish (*Danio rerio*) (*Heras et al., 2019*). DL has already been used for fish detection in the field, most of the time to automatically detect and classify fish from underwater cameras (*Alshdaifat, Talib & Osman, 2020*; *Huang et al., 2019*; *Jalal et al., 2020*; *Li et al., 2018*; *Rathi, Jain & Indu, 2018*; *Jones, Webster & Salvanes, 2021*). CNNs have been used also in fish tanks, to monitor fish farms (*Arvind et al., 2019*) and to study collective movements (*Heras et al., 2019*; *Romero-Ferrero et al., 2019*). However, to our knowledge, no one has yet applied this technology to study fish behavioural types, a growing field with vast ecological implications (*Conrad et al., 2011*; *Mittelbach, Ballew & Kjelvik, 2014*; *Smith & Blumstein, 2008*).

The application of CNNs to analyse images and videos with the aim of extracting relevant behavioural information under laboratory conditions is still in its infancy but it is not difficult to foresee the major advance that this technology can represent in the field of behavioural ecology. CNNs have been particularly successful because of their versatility and low computational cost. The Faster R-CNN (Faster region-based convolutional neural network), used in this work, is one of the most successful CNNs of the last years (*Ren et al., 2017*). It represents the third iteration of the network R-CNN: in 2014, the R-CNN was published by Ross Girshick (*Girshick et al., 2014*); the following year, it rapidly evolved into the Fast R-CNN (*Girshick, 2015*) and then into the Faster R-CNN. It was designed for multiple object detection and classification in images, and it outputs some object proposals, each one with two associated predictions: the probability to appertain to a class and the bounding box that locates the object in the image. The Faster R-CNN has been applied successfully in a wide range of fields including car detection (*Xu et al., 2017*), face detection (*Jiang & Learned-miller, 2017*) or, in medicine, to identify specific osseous landmarks from radiographs (*Sa et al., 2017*). This neural network has also been applied to ecology research such as seagrass detection (*Moniruzzaman et al., 2019*), butterfly recognition (*Zhao et al., 2019a*, *Zhao et al., 2019b*) and whales' social interactions (*Rasmussen & Širović, 2019*).

The objective of this work was to train and validate a Faster R-CNN to track fish movements during behavioural tests in experimental arenas, with the final aim to serve as an automatic tool for assaying individual differences in fish behavioural types. We applied the neural network to video recordings of the gilthead seabream, *Sparus aurata*, a relevant species for aquaculture industry. Individuals of this species were maintained in experimental behavioural arenas under controlled conditions, continuously recorded with a Raspberry Pi (https://www.raspberrypi.org/), and subjected to daily behavioural tests. We used images obtained from the videos to train and validate the Faster R-CNN for automatic fish detection and positioning. Our trained neural network, coupled with the Raspberry Pi-based recording system and applied to behavioural tests in the experimental arenas, will serve as a tool for automatically quantify individual differences in fish behavioural types and evaluate the presence of behavioural syndromes (*Castanheira et al., 2013*; *Conrad et al., 2011*; *Réale et al., 2007*).

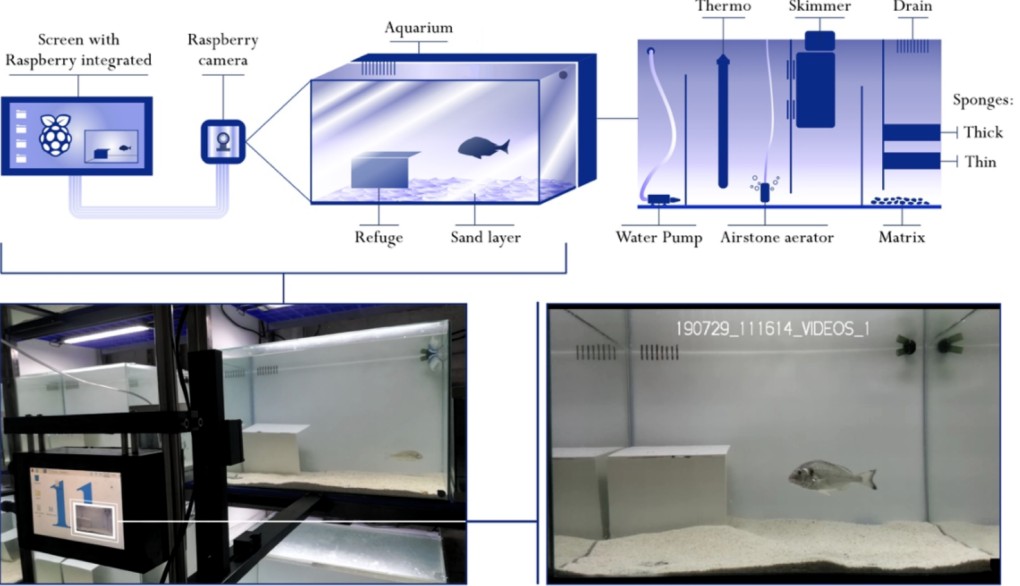

**Figure 1** **Diagram of the experimental behavioural arenas and recording system.** Behavioural arenas were enriched with a sandy layer bed and a plastic box as refuge. Every aquarium was equipped with a re-circulating system (pump), a filtering system (skimmer, two sponges and a rock matrix), a water heater for regulating the water temperature and an airstone aerator. The activity of the fish during the behavioural tests was continuously recorded by a camera controlled by a Raspberry Pi 3.

## MATERIALS & METHODS

### Experimental set-up of the behavioural arenas

The ultimate aim of this work was to provide a tool to study fish behavioural types or consistent individual differences in different contexts. We applied our neural network to video recordings of individuals of gilthead seabream maintained in isolated experimental behavioural arenas under controlled conditions while performing behavioural tests. Behavioural arenas were composed by a main aquarium ($60 \times 50 \times 40$ cm, 120 L) enriched with a shelter and a sandy bottom (one cm deep, with grains of 0.5–1.2 mm), filled with sterilized seawater (21 °C), and maintained by a filtering system composed by physical and biological filters, including a skimmer, a closed recirculation system and an aerator (Fig. 1). Within the experimental area there was a shelter that provided a place where the fish could hide and, for the purpose of this study, it was helpful to create two different conditions to evaluate the performance of the neural network: the presence or absence of the fish in the video when outside or inside the shelter.

During the experimental week, fish were continuously recorded with a camera (Raspberry Pi Camera Module V2 of 8MP) controlled by a Raspberry Pi 3 Model B+. The Raspberry was lodged in a plastic case to prevent water damage and equipped with a 120 GB USB stick to save the recordings. We set up 12 behavioural arenas in our experimental isolated room, each one with a Raspberry Pi and a camera in front of it, that permitted us to perform behavioural tests on 12 fish simultaneously (Fig. 1). In total,

we tested 108 individuals of gilthead seabream, obtaining more than 14,000 h of video recordings. The videos, recorded at 1080p with a framerate of 30 fps, were first used to obtain the frames to train and validate our neural network. Once the final version of the network was generated (see results), we used it to analyse the videos of all the tests, obtaining the position of the fish in every frame. Combining the subsequent positions, we obtained the complete fish trajectory, from which, using different metrics, we extracted relevant behavioural information.

## Collection of the individuals

Our experiment was part of a project whose goal was to study the correlation between fish behaviour and gut microbiome. More specifically, to search for consistent differences in behaviour and gut microbiome between two different strains of gilthead seabreams: individuals bred for aquaculture and wild individuals. Captive-bred individuals were provided by the Institute of Agrifood Research and Technology (Spain) in July 2019 from their annual breeding program. They were transported and housed in the Marine and Aquaculture Research Laboratory (LIMIA) in Port d'Andratx (Balearic Islands, Spain), following the standard conditions described in *Ortega, 2008*. Wild individuals were captured in the waters of Mallorca island using conventional hook-and-line gear (one single hook attached to a line with a fishing float) and transported to the LIMIA using an oxygenate tank (50 L). Before the experiments in the behavioural arenas, all the individuals were housed for 1 week in collective (5,000 L) tanks for acclimation. During the experiments, the wild seabreams were fed daily with 3 shrimps per individual while the captive-bred seabreams with 1 g of conventional pellet food (D-2 OptiBream AE 1P and D-4 OptiBream AE 2P) per individual, according to the standardized protocols for the species. All the laboratory procedures were approved by the Ethical Committee for Animal Experimentation of the University of the Balearic Islands (ref. CEEA 98/07/18), and authorised by the Animal Research Ethical Committee of the Conselleria d'Agricultura, Pesca i Alimentació and the Direcció General de Pesca i Medi Marí of the Government of the Balearic Islands.

## Behavioural quantification

For the purpose of the main project, following the five axes of fish behavioural types described in *Réale et al., 2007*, we performed five different types of tests: exploration-avoidance, aggressiveness, sociability, shyness-boldness and activity. In this study, to train and validate our neural network, we used videos from only two of the tests, activity and exploration:

— Activity: this test measured the frequency of movement in a safe and familiar environment. The individual's activity was measured with an open-field test. Using the fish trajectories obtained with the neural network, we measured (i) the time spent outside the shelter, (ii) the total time spent swimming, (iii) the average turning angle and (iv) the area of the aquarium covered by the fish.

— Exploration: this test measured the individual willingness to explore a new situation, including new habitats and new objects. It consisted in introducing a novel object (a coloured toy animal figure) in the experimental aquarium and observe the individual's

first reactions. The exploration-avoidance behaviour was quantified measuring (i) the time spent outside the shelter, (ii) the minimum distance of approach to the novel object, and (iii) the time spent near the object (in a radius of 100 pixels from the centroid of the object).

These experiments represent the two main situations in which the neural network must be able to recognize the fish. In fact, as we will discuss later, in exploration videos the presence of the novel object affects the inference of the neural network. We saw in previous trials that after having been introduced in the behavioural arenas, it took three days for gilthead seabreams to display normal behaviour and feeding habits. For this reason, before starting the experiments, every fish was acclimated for 3 days in the aquarium. Each test lasted 1 h (except for activity that lasted 2 h) and was carried out once a day for 4 consecutive days. The order of the test was randomized to avoid within-day temporal influence on behaviour.

## Faster R-CNN functioning

We used the pre-trained neural network Faster R-CNN Inception-V2 COCO. This neural network is a Faster R-CNN with architectural modification (Inception-V2) at the level of single layers and pre-trained with the COCO dataset. The Inception architecture was an important milestone in the development of CNNs classifiers, because it boosted their performance by making the layers wider rather than deeper and using kernels of different sizes (*Szegedy et al., 2015*; *Szegedy et al., 2016*; *Raj, 2018*). The Faster R-CNN has been designed to work with images, for this reason, we extracted 3,700 independent frames (one per second) for each hour of video recording. The input provided to the Faster R-CNN is an image that will be processed in sequence by several dynamic parts (Fig. 2). First, a pre-trained CNN creates a first feature map from the image. The pre-trained network was trained, previous to our use, on a big dataset and has already calibrated the weights to extract relevant features according to the task. The usage of a pre-trained network, that falls into a field called transfer learning (*Goodfellow, Bengio & Courville, 2016*), allows for the use of a small dataset for the training, reducing substantially our effort (*Hendrycks, Lee & Mazeika, 2019*). Our neural network was pretrained with COCO dataset that is composed of 123,287 images of different types of objects (http://cocodataset.org/). The second part of the Faster R-CNN pipeline is the Region Proposal Network (RPN), that chooses from a fixed set of anchors (boxes distributed uniformly on the image with different sizes and ratios) the one that may contain an object. This first prediction is then used as starting points to predict the bounding boxes (Fig. 2). Then, the Region of Interest Pooling (RoIP) extracts fixed-size feature maps from the proposals of the RPN and sends them to the last part, the R-CNN (Fig. 2). Finally, this CNN classifies the objects proposed and adjusts their bounding boxes' coordinates for a better fit (Fig. 2). At the end, we obtain bounding boxes that identify where the target objects are in the initial image, each one with a class and a percentage of reliability for the classification (Fig. 3) (*Rey, 2018*). Because in our experiments there was only one fish at a time and it was always of the same species (see below), we set the neural network to assign only one class (gilthead seabream) and to give us only the most reliable object proposal. The neural network we used can be accessed at (https://github.com/tensorflow/models/tree/master/research/object_detection). To manage
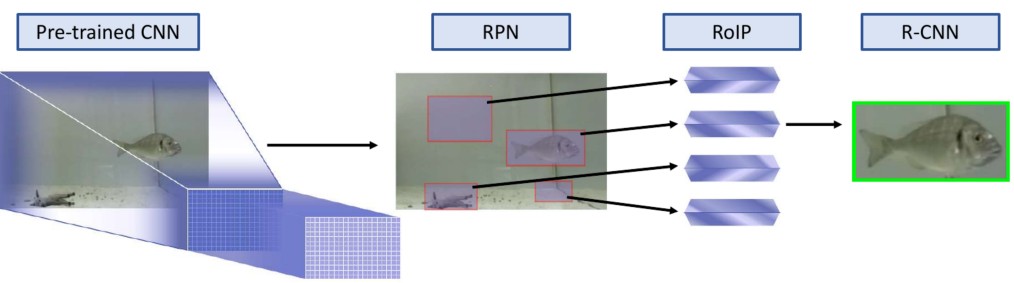

**Figure 2 The four-step Faster R-CNN pipeline applied to our study case.** (1) A pre-trained CNN creates a first feature map from the image. (2) The Region Proposal Network (RPN) proposes some region that may contain an object, to use it as starting points to predict the final bounding boxes. (3) The Region of Interest Pooling (RoIP) extracts fixed-size feature maps from the proposals of the RPN. (4) The final region-based CNN (R-CNN) classifies the objects proposed and adjusts their bounding boxes. At the end, the output is a bounding box that identifies where the target object is in the initial image, with a class and a percentage of reliability of the associated classification.

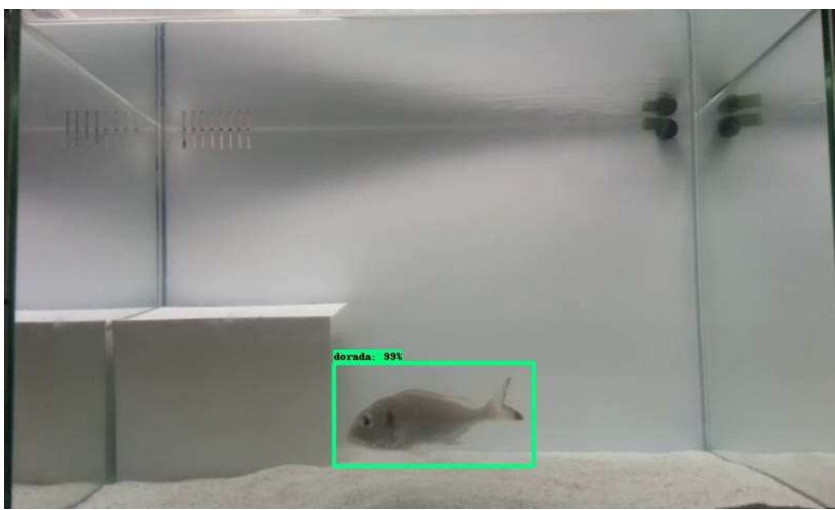

**Figure 3 Example of the final output of the Faster R-CNN applied to the identification of the gilthead seabream,** *Sparus aurata.* The green rectangle is the bounding box, with the associated class (dorada, Spanish for gilthead seabream) and the classification probability of the prediction (99%). The bottom-left corner of the image shows the experimental set up of the behavioural arena, with the sandy bottom and the shelter.

it, we used the open-source library Tensorflow (https://www.tensorflow.org/) with Python 3.

## Image dataset and training of the Faster R-CNN

To create the training dataset, we selected 60 one-hour videos from the over 14,000 h of recordings. In this selection, to enable the algorithm to work in different conditions, we ensured to include all the possible variability in the quality of the recordings, such as different light conditions (depending on the position of the aquarium in the room),

visibility (water on the external side of the aquarium may produce tarnished images), and video disturbances. Covering all the possible video conditions in the training dataset is fundamental to enable the algorithm to function in as many situations as possible. From each one of these videos, we randomly extracted a maximum of 250 images (frames, .JPG format, 1,280 × 768 px). For the same reason as the video selection, these images covered most of the possible postures and positions the gilthead seabream could display. Then, we annotated the images with the free software LabelImg (https://github.com/tzutalin/labelImg). We created two groups: the "train" group for the actual training of the neural network containing 80% of images, and the "test" group for testing after training, containing 20% of images.

We performed four training sessions, three increasing the number of images and a fourth using an augmentation technique. After every session, we validated the functioning of the neural network (see next section), to evaluate the effect of increasing the number of training images and applying the augmentation on the accuracy of the predictions. For the first training session, we used a total of 6,000 images of gilthead seabreams randomly sampled from different videos. In the second one, we added 4,000 images containing, in addition to gilthead seabreams, the novel objects introduced in the exploration tests, to train the network to not recognize the novel object as fish. In the third training session, we added other 4,000 images to try to further improve the accuracy of the neural network, reaching a total of 14,000 images. The number of images used for the training is a very important parameter in deep learning: too few images would not allow the neural network to learn how to perform its task (underfitting), while too many images would make it too "adapted" to the training dataset, reducing its ability in detecting an object in never-seen-before images (overfitting) (*Shahinfar, Meek & Falzon, 2020*). Moreover, looking at how the neural network changes its predictions in relation to the number of images can provide us with important information on how it works and how to improve it.

In the last training session, we added an augmentation to the 14,000 images used for training in the previous sessions. Image data augmentation is a method used to artificially expand the size of a training dataset by creating modified versions of the original images (*Perez & Wang, 2017*). We set the neural network to use five types of modifications, that may be useful in detecting a fish: Random Horizontal Flip, creates mirror images starting from random points along the horizontal line; Random Distort Colour, randomly changes the colours of the image; Random Adjust Brightness, modifies its brightness; Random Crop Image and SSD Random Crop, crops a part of an image with its bounding box. Data augmentation is a very common technique in image analysis with deep learning, and not only it reduces the training effort, but it proved to substantially increase the accuracy of the model (*Wong et al., 2016*).

The hyperparameters of our neural network have already been fine-tuned for the Oxford-IIIT Pets Dataset (https://www.robots.ox.ac.uk/~vgg/data/pets/). We used the stochastic gradient descent (batch size = 1) with a stable learning rate of 0.0002 for 187,500 steps and a momentum optimizer of 0.9. The training was carried out on an Intel(R) Xeon(R) W-2155 server (CPU 3.30 GHz, 64 GB of RAM DDR4 and GPU NVIDIA Quadro GV100).

## Validation of the trained neural network

To evaluate the performance of the Faster R-CNN in detecting and positioning individuals of gilthead seabream, we performed two different validations, one for the detection/classification task and another for the positioning (positioning error in distance). The two validations consisted in analysing the same images manually and by using our neural network and comparing the results. The images used for the validation were new to the algorithm, they were not previously used for training.

To validate the detection/classification task, we randomly sampled 14 one-hour videos (11 from activity and three from exploration) and validated them after every one of the four training sessions (see previous section). The manual analysis for this validation was done by visualizing the videos and recording the presence or absence (outside or inside the shelter) of the fish. We generated a time series of the presence and absence (1 and 0) of the fish (Fig. 4). In parallel, we analysed the same videos using the Faster R-CNN. Even if the videos were recorded at 30 fps, the neural network analysed only one frame every second (for a total of 51,800 frames in the 14 videos), providing for every one of them one value: the probability of one detected object (here, only the one with the highest probability) to appertain to a class (in our case, the only class is to be a gilthead seabream). We can interpret this value, ranging from 0 to 1, as the probability of the presence of a fish in an image. We set a threshold from which to consider a fish present, transforming the continuous output into a discrete one, obtaining a time series composed only by 0 and 1, comparable with the manual annotation. For every validation we used five different thresholds (0.5, 0.6, 0.7, 0.8 and 0.9), to obtain the most accurate prediction and to better understand the probability distribution of the output. The overall validation process generated two time series per video (manual vs. Faster R-CNN detection/classification) and we compared them to calculate the matching percentage. For every video, we compared first the whole time series, and then the time series separating the parts in which the fish was present, according to the manual analysis, from the parts in which it was absent (Fig. 4). In this way, we could easily visualize if the algorithm had more problems in predicting the presence (false negatives) or the absence (false positives) of a fish. We provide the mean, the standard deviation [confidence interval] and the root-mean-square error (RMSE), as proxy of error of the CNN (*Chai & Draxler, 2014*).

In addition to the classification probability, for every image the neural network gives us a positioning output, that are the coordinates (in pixels) of the centroid of the bounding box that localizes the fish in the image. We validated this output only for the best training step in detecting the fish. From the 14 validation videos (51,800 images), we randomly sampled 1,085 images in which the fish was present, according to the manual analysis. To mitigate the autocorrelation, the number of images sampled from each video was proportioned to the portion of the video in which the fish was present. Then, using LabelImg, we manually drew the bounding boxes in the images to obtain and extract its coordinates. From these coordinates, we calculated the centroids and we compared them with the outputs of the Faster R-CNN to calculate the error of positioning (distance in pixels), between the two points. In this case, to compute the percentage of success, we used as threshold the average length of the sides of all the bounding boxes drew manually in

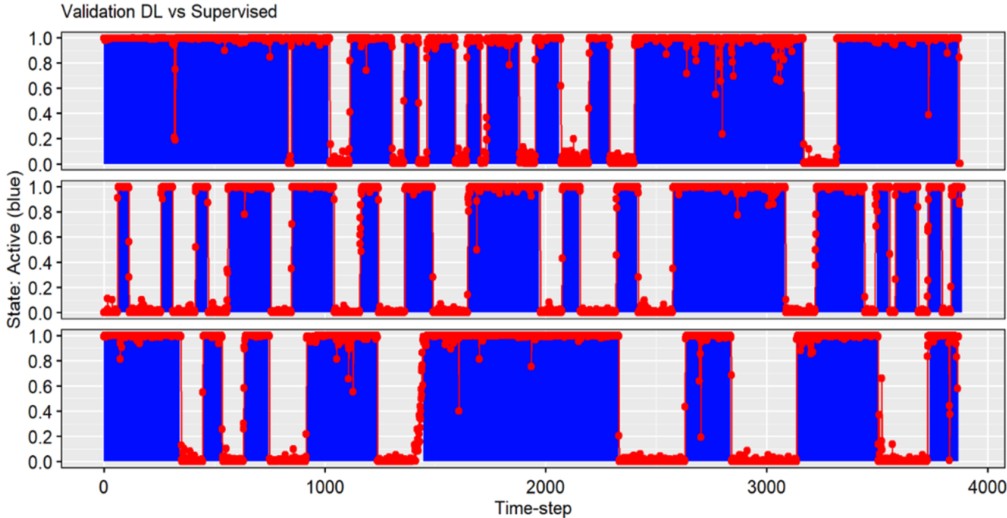

**Figure 4** **Temporal series of three video samples.** The *x*-axis represents time in seconds, and the *y*-axis the probability of detection of the fish. In blue bars, the human-based manual analysis (supervised) being 1 when the fish was present (outside the shelter) and 0 when it was absent (inside the shelter). The scatterplot in red represents the probability of a fish being present calculated by the deep learning (DL) algorithm.

all the frames. As for the classification task, we provide the mean, the standard deviation [confidence interval] and the root-mean-square error (RMSE) for the positioning error. All the processes of this validation were carried out in R (https://www.r-project.org/) and in Python 3 (https://www.python.org/).

## RESULTS

For the detection/classification output, using a higher number of images for the training and the application of the augmentation technique increased substantially the accuracy of our neural network (Table 1). The overall percentage of correct predictions, calculated as the average of the correct predictions for the five different thresholds, increased from 75.86% ± 25.77% in the first training step (6,000 images) to 90.49% ± 11.45% in the fourth one (14,000 images plus data augmentation). Using different thresholds did not have an important effect on the predictions, but in general low threshold values (from 0.5 to 0.7) resulted in better predictions. The best threshold was 0.5. Using this threshold, the increase in accuracy was more evident, from 78.2% ± 24.73% in the first training step to 92.79% ± 6.78% in the fourth (Table 1 and Fig. 5). We found differences in the results of the validation between the parts of the videos where the fish was present and the parts where the fish was hidden in its shelter. The neural network was more accurate when the fish was present (from 72.78% to 83.70% with the successive training and augmentation rounds) than when predicting its absence (from 61.43% to 65.45%) (Table 1).

By increasing the number of images, the accuracy of predictions in the activity videos did not change substantially (Fig. 5). The accuracy for the exploration videos improved

Table 1  **Results of the validations at the four training steps of our Faster R-CNN.** The table shows the accuracy (as the mean % of success and its standard deviation, SD) for the activity test (ACT), the exploration test (EXP) and the combination of both tests (Total) at different thresholds (from 0,5 to 0,9) adopted to consider the fish present or absent. The first row of every type of validation (6,000, 10,000, 14,000 and Augmentation) contains the mean values of the results at every threshold value for each one of the four training steps. For the whole tests (Total) the table also shows the confidence interval and the RMSE. The values in bold are the results of the validation using a threshold of 0,5, that is the one we chose to use in our experiments.

| Type of validation | ACT | | EXP | | Total | | | |
|---|---|---|---|---|---|---|---|---|
| | Mean % success | SD | Mean % success | SD | Mean % success | SD | Confidence interval | RMSE |
| **6000** | 80.34 | 26.53 | 59.45 | 13.80 | 75.86 | 25.77 | 69.83–81.9 | 47.30 |
| **0.5** | **83.11** | **24.67** | **60.20** | **17.69** | **78.20** | **24.73** | **65.25–91.15** | **41.92** |
| 0.6 | 81.87 | 26.34 | 59.94 | 17.10 | 77.17 | 25.81 | 63.65–90.69 | 43.06 |
| 0.7 | 80.65 | 27.62 | 59.57 | 16.37 | 76.13 | 26.62 | 62.19–90.08 | 44.06 |
| 0.8 | 79.15 | 28.86 | 59.21 | 15.77 | 74.88 | 27.41 | 60.52–89.23 | 52.51 |
| 0.9 | 76.93 | 29.62 | 58.33 | 14.41 | 72.94 | 27.74 | 58.41–87.47 | 53.62 |
| **10000** | 79.21 | 28.47 | 93.32 | 7.62 | 82.23 | 26.08 | 76.12–88.34 | 41.24 |
| **0.5** | **80.69** | **29.61** | **93.50** | **7.47** | **83.44** | **26.70** | **69.45–97.42** | **40.63** |
| 0.6 | 80.16 | 29.58 | 93.58 | 7.95 | 83.04 | 26.75 | 69.03–97.05 | 40.82 |
| 0.7 | 79.45 | 29.52 | 93.47 | 8.79 | 82.45 | 26.79 | 68.42–96.49 | 41.09 |
| 0.8 | 78.59 | 29.51 | 93.28 | 9.58 | 81.74 | 26.89 | 67.65–95.82 | 41.46 |
| 0.9 | 77.14 | 29.59 | 92.77 | 10.83 | 80.49 | 27.12 | 66.28–94.7 | 42.17 |
| **14000** | 86.84 | 22.45 | 93.65 | 6.48 | 88.29 | 20.30 | 83.54–93.05 | 52.98 |
| **0.5** | **88.55** | **21.85** | **93.75** | **6.27** | **89.67** | **19.58** | **79.41–99.92** | **47.82** |
| 0.6 | 88.20 | 21.79 | 93.76 | 6.39 | 89.39 | 19.55 | 79.15–99.63 | 47.87 |
| 0.7 | 87.53 | 21.90 | 93.69 | 6.69 | 88.85 | 19.70 | 78.53–99.17 | 47.93 |
| 0.8 | 86.23 | 22.64 | 93.64 | 7.09 | 87.82 | 20.43 | 77.11–98.52 | 48.01 |
| 0.9 | 83.67 | 25.62 | 93.39 | 8.25 | 85.75 | 23.23 | 73.58–97.92 | 48.21 |
| **Augmentation** | 90.41 | 12.40 | 90.77 | 7.32 | 90.49 | 11.45 | 87.8–93.17 | 22.48 |
| **0.5** | **93.39** | **6.86** | **90.61** | **7.38** | **92.79** | **6.78** | **89.24–96.34** | **28.44** |
| 0.6 | 92.84 | 6.78 | 90.73 | 7.64 | 92.39 | 6.72 | 88.87–95.91 | 28.53 |
| 0.7 | 91.71 | 8.04 | 90.76 | 8.16 | 91.51 | 7.75 | 87.45–95.57 | 11.31 |
| 0.8 | 89.36 | 12.25 | 90.90 | 8.88 | 89.69 | 11.32 | 83.76–95.61 | 15.01 |
| 0.9 | 84.75 | 21.62 | 90.84 | 10.83 | 86.05 | 19.60 | 75.78–96.32 | 23.48 |

substantially between the first and the second training steps, from 59.42 ± 13.8% to 93.32 ±7.62%. We noticed two activity videos that remained with low accuracy values, one around 50% and the other one close to 0%, even increasing the number of images (Fig. 5). These videos improved applying the augmentation technique, reaching values higher than 80% at any threshold (Table 1). In general, for all the other videos, the augmentation did not have an important effect on the accuracy of the model and is some cases it even resulted in slightly worse predictions (Fig. 6). The most accurate training step was the last one, with 14,000 images, with data augmentation and using a threshold of 0.5. The neural network reached an accuracy in the classification of 92.79% ± 6.78% [89.24–96.34] and a RMSE of 28.44. It means that from 51,800 analysed images, in 47,328 images was successfully assessed the absence or presence of the fish.

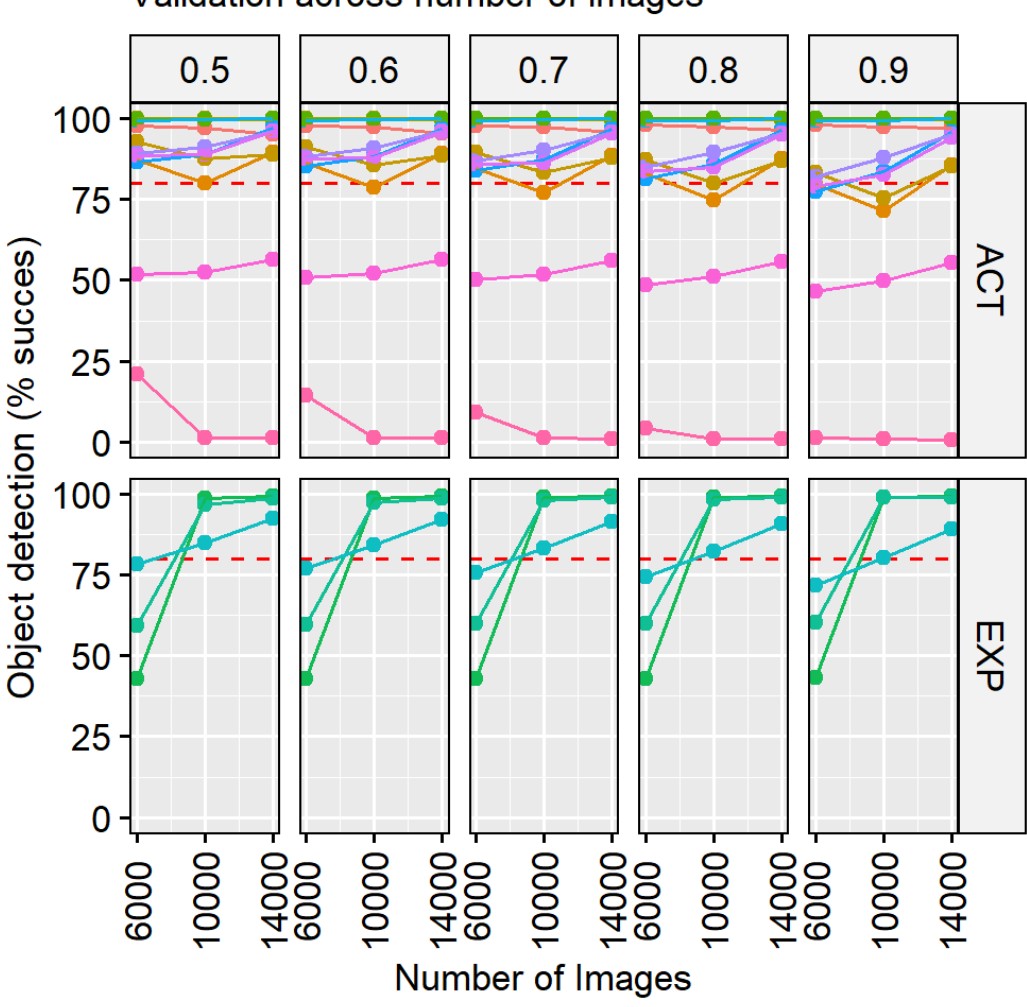

**Figure 5** **Validation results increasing the number of images used for training.** The diagram shows the number of used training images (*x* axis), the percentage of object detection success (y axis), the type of experiment activity (ACT) and exploration (EXP) and the threshold used (headings). An increasing number of images was used on the first three training phases (6,000, 4,000 and 4,000 for a total of 14,000). Each colour represents a video, and the red dashed line highlights the percentage (80%) of success.

With this last version of the neural network, we validated the positioning output by calculating the error (distance between real and estimated centroids of the target object) of the predictions in 1,085 frames (Fig. 7). Most of the errors resulted to be lower than 25 pixels (based on an image of 1,280 × 768 px) with a mean and standard deviation of 10.25 ± 61.59 pixels [6.59–13.91] and RMSE of 62.42 pixels, therefore, below the average length of the sides of the bounding boxes that we used as threshold. With this neural network we reached a 98.9% accuracy of correct positioning of the fish (Fig. 7).

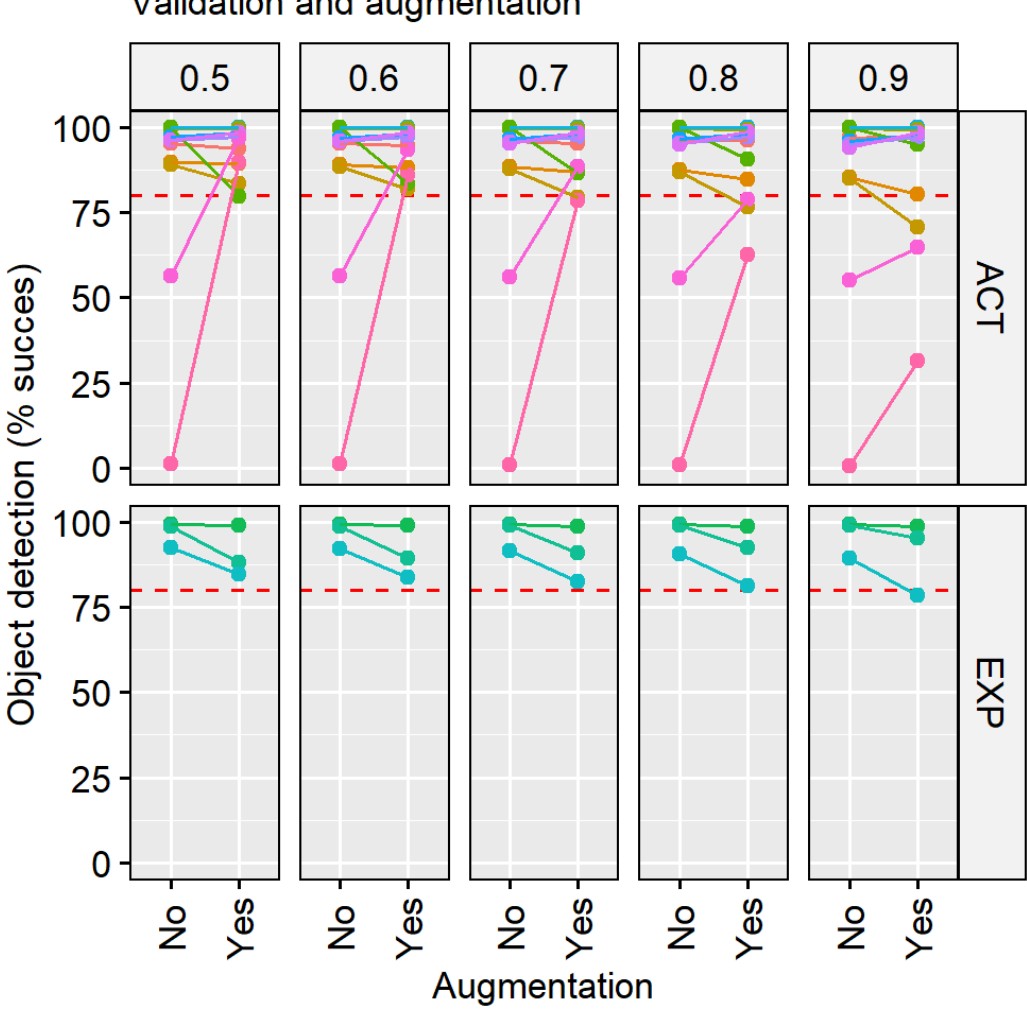

**Figure 6** **Validation results before and after have used data augmentation.** The diagram shows the performance of our neural network before and after applying data augmentation (*x* axis) expressed as percentage of object detection success (*y* axis) by type of experiment activity (ACT) and exploration (EXP) and threshold used (headings). Each colour represents a video, and the red dashed line represents the percentage (80%) of success.

## DISCUSSION

Here, we have successfully adapted and trained the Faster R-CNN to track recorded gilthead seabreams to study fish inter-individual variation in behavioural types. Compared to human-led analysis, that can be time-consuming, the usage of a trained neural network significantly reduced the time effort needed. In order to obtain a neural network to achieve this task, we trained a pre-trained Faster R-CNN to recognize a gilthead seabream in images, using a total of 14,000 manually labelled images and an augmentation technique to artificially expand the dataset. We trained the network in four steps, increasing the number of images during the first three (6,000, 10,000 and 14,000 images) and adding the

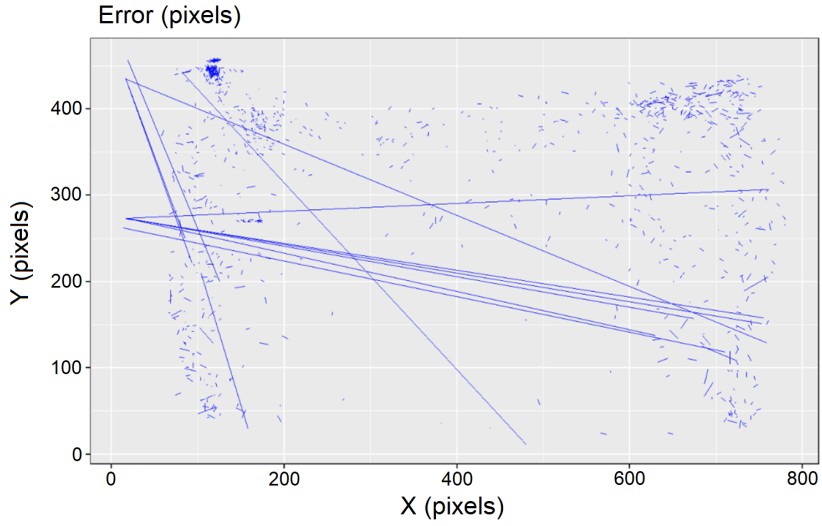

**Figure 7 Validation of the positioning output.** The blue lines represent the coordinate differences between the manually labelled bounding boxes and the predictions of the algorithm for all the validation images, thus representing the positioning error (the longer the line the greater the error). The plot represents the area of the experimental behavioural arena.

augmentation technique in the fourth. Then, we validated the functioning of the network for the two tasks, classification/detection and positioning, comparing a human-led analysis and a deep learning-based one. We found that the network trained with the maximum number of images and with the augmentation technique reached the best accuracy, correctly predicting the presence of a fish the vast majority of times (92,79% ± 6.8% of success), and rightfully localizing its position a 98,89% of the times. We therefore provide a new automatic tool to detect and analyse the behaviour of this important commercial fish species.

Before applying deep learning algorithms to our video dataset, we used classic computer vision (CV) methods (thresholding and background subtraction) to track fish in the behavioural arenas (*Lee, Wu & Guo, 2010*). The main difficulties we encountered were that the CV algorithm frequently lost its target when it moved fast (appearing blurred in the frames) and when it was near the bottom, where the clear abdomen resembled the colour of the sand, producing positioning errors. Nevertheless, this was not the main reason for switching to a deep learning approach, as the efficiency of a well-implemented traditional CV tracking algorithm was not questioned. However, the following set of experimental trails in the behavioural arenas required for another fish species (*Xyrichtys novacula*), applying changing light conditions (experiments on circadian rhythms and chronotypes require day and night light cycles) and presenting new objects (*Martorell-Barceló et al., 2021*). Under these circumstances, we decided it was more advantageous for our research to train another neural network than having to manually readjust a CV algorithm. Since in DL the feature selection and extraction is automatic in order to train a new neural network we only need to prepare a sufficiently representative new dataset: for this study, labelling

14000 images took less than 8 h, without the need of specialised personnel. For this added to the fact that we were not limited by a lack of data or computational power (the two main limits of DL), we decided to take the deep learning approach.

An interesting aspect of the validation we performed in this work concerns the different performance of our neural network when the fish was present or absent (hidden in the shelter). At all stages of training, we found that the algorithm is more accurate when the fish is present than when it is absent. This can be caused by the variability of the information we are giving to the neural network regarding the object we want to detect and the image background. When we selected new images for the training, we always tried to cover all the possible postures a fish can display, to make the algorithm able to recognize it in as many situations as possible (*e.g.*, showing only its head outside of the shelter). In this manner, the fish is more variable than the background, which is almost always the same. This means that we are giving the network more information about what it is a fish than about what it is background, leading to a greater improvement of the accuracy when the fish is present. To fix this unbalance, it would be necessary to train the neural network with new images containing some "variation" in the background or at least to train the algorithm without repetitive frames taken from the same videos. We decided not to correct this unbalance since the accuracy achieved by the neural network was enough to fulfil our needs regarding the data collection from the behavioural experiments.

When detecting an object, the probability threshold has a strong impact on improving the accuracy of a neural network. The threshold is the value we set to establish the minimum probability upon which the algorithm considers a fish as present. It is an important parameter to set because, depending on the model and on the distributions of the errors, it can lead to large effects on the results (*Liu et al., 2005*). In our work, we saw that the choice of the best threshold depends directly on the percentage of the presence of a fish in a video. In fact, videos with high presence of a fish worked better at low thresholds, which permits to filter out the majority of the false negatives. On the other hand, for videos where the fish is absent most of the time, a higher threshold is preferred. Being the percentage of the presence of a fish in a video highly variable and unpredictable, and assuming that its distribution follows the central limit theorem, we resolved this problem using a threshold of 0.5, which in fact generated the best performance of the network. For the positioning task, the output is a set of coordinates in pixels (not a probability), thus in this case the threshold gives us a final accuracy percentage, being less important than in the previous task. Due to the distribution of the errors of the positioning outputs (the vast majority are less than 25 pixels and a fish measured 200 pixels), the choice of the threshold here has less importance on the result of the validation, given that we chose a reasonable one.

The Faster R-CNN was even better at positioning than classification, with a positioning error of $10.25 \pm 61.59$ pixels (in an image of the aquarium of $1,280 \times 768$ px). These differences are consistent with the conclusions of *Wu et al. (2020)* that compared the functioning of two different types of neural network for the two tasks. Briefly, they found that neural networks that start with fully connected layers (fc-head) are more robust in classifying objects, while networks that start with convolutional layers (conv-head), like the Faster R-CNN, are more suitable for the localization (regression) task. This may be
due to a higher spatial sensitiveness of the fully connected layers, that have more capability to distinguish a complete object from parts of it, using different parameters for different parts of a proposal. On the contrary, convolutional layers share the same kernel for the whole proposal and, thus, are more robust to regress the whole object (*Wu et al., 2020*). In all cases, our neural network provided acceptable values for both the detection and the positioning of the fish after having been trained with 14,000 images and with data augmentation.

Dividing the training process in different steps allowed us not only to monitor the performance of the neural network, avoiding underfitting and overfitting (*Jabbar & Khan, 2015*), but also to resolve other problems specific to our experimental conditions. In fact, in the exploration experiments the algorithm recognized the "novel object" as a fish when the fish was not present (Fig. 8). We fixed this problem with the second training step, during which we fed the network with 4,000 images all containing both a fish and a toy, to emphasize to the algorithm that the toy was not of our interest (Fig. 8). The correct predictions of the exploration videos increased substantially from the first to the second validation (Fig. 5). In general, the increase in number of training images led on average to better predictions (*Barbedo, 2018*), and image data augmentation permitted us to resolve another problem: the network was not able to recognize a fish when it was partially cut out of an image. Image data augmentation is a common technique that proved to be very efficient in improving the precision of a neural network (*Perez & Wang, 2017*). Here, we used five augmentation transformations: Random Horizontal Flip, Random Distort Color, Random Adjust Brightness, Random Crop Image and the SSD Random Crop. Data augmentation solved all the issues in which the network was not able to recognize a fish because it was cut out of the image or halfway into the shelter (Fig. 8). Similar approaches applying data augmentation have led similar results improving the accuracy of neural networks in detecting objects (*Huang et al., 2019*), suggesting data augmentation as a useful technique to improve fish deep learning algorithms without having to increase the sample size of training images.

The last version of the trained Faster R-CNN permitted us to automatically analyse 1,550 videos (more than 2,000 h of recordings) and to generate the raw material to estimate behavioural types in our studied species. We were able to obtain the position of the fish every second (one every 30 frames) for the entire duration of the experiments (Fig. 9). From these positions, we extracted metrics to evaluate the overall activity, such as the number of movements carried out (locomotion), the times spent inside the shelter or moving, the distance travelled, the area of the behavioural arena used and the angles of the fish trajectory. Regarding the novel object test, the output generated allowed us to compute classical exploration metrics such as the minimum distance to the novel object or the time spent interacting with it (*Conrad et al., 2011*).

The natural next step to improve our system is to run the detection algorithm directly on the Raspberry Pi in real time, to obtain fish trajectories without having to store the videos or using other computers for the analysis. A Raspberry Pi 3 (1 GB of RAM) can run our model at almost 0.05 fps (analysing one image every 20 s) without hardware add-ons. To speed up the inference, we can add an external GPU (Graphics Processing Unit) to the
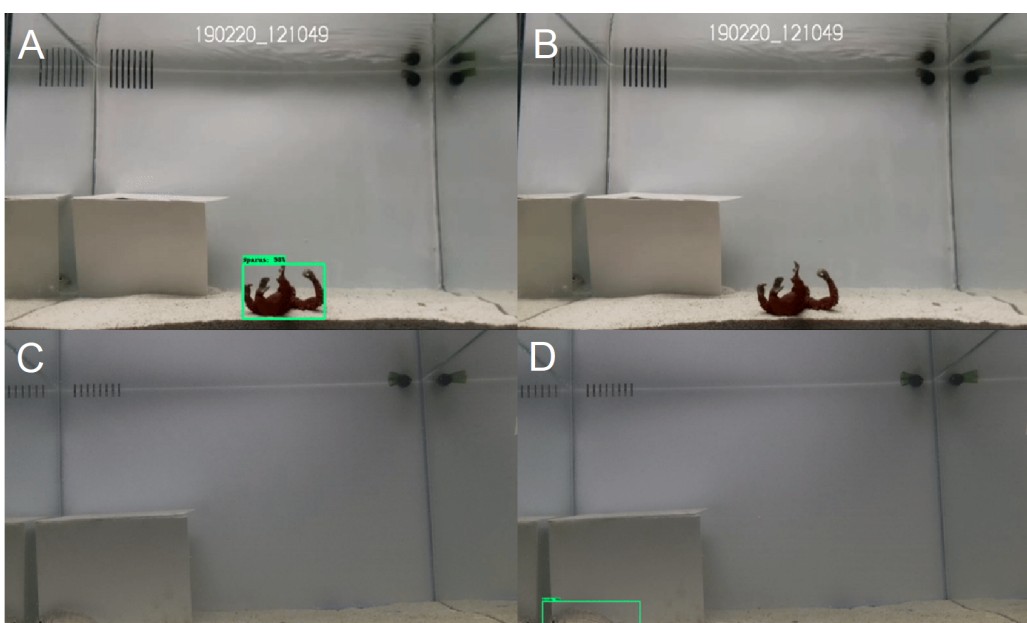

**Figure 8 Examples of the two main errors of the neural network that were solved during the training sessions.** (A) Image where a novel object of the exploration tests was detected as a fish (Sparus). (B) Same image analysed after having solved the error training the neural network with images containing novel objects. (C) Image with a non-detected fish on the bottom-left corner. (D) Same image analysed after having solved the problem using data augmentation.

Raspberry Pi: preliminary trials using MOVIDUS (*Pester & Schrittesser, 2019*) allowed us to detect and track in real-time the fish with a sufficient framerate. Our system can be easily adapted (retraining the neural network) to recognize other species, such as, for example, the pearly razorfish (*Xyrichtys novacula*) and the spiny lobster (*Palinurus elephas*), both in laboratory and in the wild using underwater cameras. Another fascinating goal we would like to achieve in improving our neural network is to automatically estimate the posture of the fish, training the neural network to recognize relevant parts of the fish like, for example, the eye (*Mathis et al., 2018*). This may provide us with information about the orientation and the field of view of the fish in the behavioural arena, giving us more precise metrics for interactions with objects or other individuals.

## CONCLUSIONS

In this work, we trained a Faster R-CNN to automatically track gilthead seabreams from video data during behavioural assays in experimental arenas. The neural network was able to correctly predict the presence of a fish for 92.79% of the times and with a positioning accuracy of 98.89%. Our trained neural network, coupled with the Raspberry Pi recording system, proved to be a powerful tool to adopt in behavioural ecology, able to automatize with high accuracy the analysis of behavioural video data.

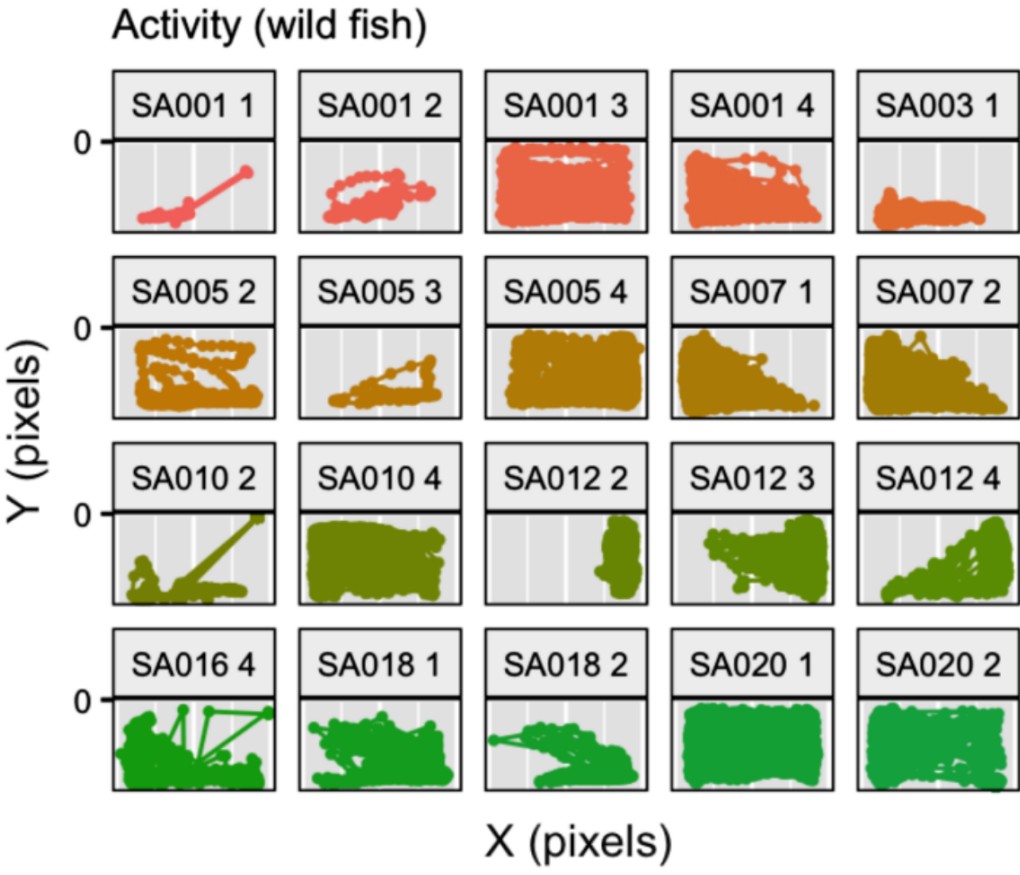

**Figure 9 Examples of results obtained from the activity experiment.** We can see the trajectories of the fish on 2-dimensional planes (representing the front view of the experimental arena) for different individuals. Each plot represents a two hours activity test. The heading of each plot corresponds to the individual identification number (starting with SA, *Sparus aurata* and followed by a three-digit individual identifier) and the number of the test (the last number).

# ACKNOWLEDGEMENTS

We thank all the volunteer anglers and the Marina of Porto Cristo for the capture of the wild individuals used to perform the experiments, and Dra Dolors Furones and Ms. Magda Monllaó from the Institute of Agrifood Research and Technology (IRTA, Spain) for providing artificially reared individuals. We also thank the head of the LIMIA Dr. Amalia Gran and Ms. Elena Pastor for their help in the experiments. This work is a contribution from the joint research unit IMEDEA-LIMIA.

## Funding

This project was funded by the research project FISHOBES (grant no. CTM2017-91490-EXP) funded by the Spanish Ministry of Science and Innovation (MICINN). Marco

Signaroli was suppoerted by a ''Ayudas para contratos predoctorales'' (grant no. PRE2020-095580) funded by MCIN/AEI /10.13039/501100011033 and the FSE ''invierte en tu futuro''. Josep Alós received funding from a Ramon y Cajal Grant (grant no. RYC2018-024488-I), the CLOCKS I+D+I project (grant no. PID2019-104940GA-I00) and JSATS PIE project (grant no. PIE202030E002) funded by MCIN/AEI/10.13039/501100011033 and the FSE ''invierte en tu futuro''. There was no additional external funding received for this study. The funders had no role in study design, data collection and analysis, decision to publish, or preparation of the manuscript.

## Grant Disclosures

The following grant information was disclosed by the authors:
FISHOBES: CTM2017-91490-EXP.
Spanish Ministry of Science and Innovation (MICINN).
Ayudas para contratos predoctorales: PRE2020-095580.
MCIN/AEI: /10.13039/501100011033.
FSE ''invierte en tu futuro''.
Ramon y Cajal Grant: RYC2018-024488-I.
CLOCKS I+D+I project: PID2019-104940GA-I00.
JSATS PIE: PIE202030E002, MCIN/AEI/10.13039/501100011033.
FSE ''invierte en tu futuro''.

## Competing Interests

The authors declare there are no competing interests.

## Author Contributions

- Marco Signaroli performed the experiments, analyzed the data, prepared figures and/or tables, and approved the final draft.
- Arancha Lana and Josep Alós conceived and designed the experiments, performed the experiments, analyzed the data, authored or reviewed drafts of the paper, and approved the final draft.
- Martina Martorell-Barceló conceived and designed the experiments, performed the experiments, authored or reviewed drafts of the paper, and approved the final draft.
- Javier Sanllehi performed the experiments, prepared figures and/or tables, and approved the final draft.
- Margarida Barcelo-Serra and Eneko Aspillaga analyzed the data, authored or reviewed drafts of the paper, and approved the final draft.
- Júlia Mulet performed the experiments, authored or reviewed drafts of the paper, and approved the final draft.

## Animal Ethics

The following information was supplied relating to ethical approvals (*i.e.*, approving body and any reference numbers):

This study was positively evaluated by the Ethical Committee for Animal Experimentation of the University of the Balearic Islands (ref. CEEA 98/07/18), and

authorised by the Animal Research Ethical Committee of the Conselleria d'Agricultura, Pesca i Alimentació and the Direcció General de Pesca i Medi Marí of the Government of the Balearic Islands.

## Data Availability

The raw data are available in the Supplementary Files.

## Supplemental Information

Supplemental information for this article can be found online at http://dx.doi.org/10.7717/peerj.13396#supplemental-information.

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
