# Peer review of "Measuring inter-individual differences in behavioural types of gilthead seabreams in the laboratory using deep learning"

_PeerJ, doi:10.7717/peerj.13396_

## Round 0.1 · original submission · Major Revisions

Your manuscript has to be improved because there are some major concerns. More details have to be provided in the materials and methods section and some more data could be included as supplementary figures (e.g. shows the different augmentations or potentially difficult situations for the CNN).

Reviewer 1 ·

Basic reporting

All basic reporting is good, except for clarity in some cases, which tend to be linked to methodological reporting. See Expt design section for more detailed comments and suggestions

Experimental design

I think the research questions and design is good. However the methods are not sufficiently described detail.

I mention specific issues in the general comment box below, but please focus on the behavioural assay procedure, the arena setup, the parameterisation of the software, and the analysis to compare the manual vs software scored videos. I also found some of the interpretation and claims seem to be overinflated or not adequately justified - and this may be linked to the lack of clarity in reporting of the methods. One example which highlights both major issues is line 46, and lines 192-195. Line 46 suggests the writing suggests that this ms details a novel raspberry pi recording system, but the readers is given just 3 lines of information about the system in the methods (192-195), citing a separate paper (from different authors). I am left wondering how it is novel, and what it actually is, there is certainly not enough information to be reproducible. There are similar issues in other areas
.
I also found that the method section was a bit hard to follow and it will benefit from re-organisation. There are sections that deal with the fish subjects, the behavioural experiments, the software setup and the scoring, manual scoring, and the comparative analysis but they were not in a clear order and sometimes mixed together.

Other Important issues
Provision of additional information on the occasions where there were large errors, or mismatch between video and human scoring, would be hugely useful. I would recommend providing some additional data if possible, for example, the number of such errors of certain magnitude (especially for presence/absence scores). Maybe also provide some information on what caused these errors and what steps could be taken to address them.

Discuss the very basic shapes/ structures used in your assay, e.g. the shelter used in this study is a simple box shape. How might this system be impacted other commonly used forms of shelter? I think this is a limitation of the study given the arguments used in the introduction framed the research question around addressing the issues surrounding the importance of this tool in aiding tracking and scoring in semi-natural settings (rather than plain white arenas).

Non-essential general suggestions - that may be useful in [providing a large benefit to readers and usefulness with relatively little extra work (you have already done so much of the hard work).
1. Add a glossary of terms used in the DL and tracking section, with some details as to why they are important.
2. Discuss some of the practical issues encountered in running and using the software and solutions you found/ helpful resources.

Validity of the findings

As mentioned above the research question is useful, and the data seem robust. Some of the conclusions are overstated and I would recommend revising these, or highlighting the justification for the claims (see specific comments below).

Additional comments

I think this is a good manuscript that provides a useful case study and examination into the behavioural tracking of fish using automatic image processing software. Behavioural scoring technologies are going to become more relevant as they mature and become increasingly widely used and this study has potential to be very useful for many researchers that wish to use such systems in the future. Currently I would however recommend significant revisions, with the major issue being that there is insufficient methodological detail at several levels.

Specific issues and comments:

Line 46 and elsewhere if you are arguing the raspberry pi setup you used for recording is novel, then share far more details of that, how it was setup, the code you used etc, the costs. And/ or point to an open-source location or paper detailing the technical details of your system. It is not really the focus of this manuscript.
Line 66: Not clear what you are referring to here please expand/ add details to clarify ‘interpersonal differences’ here.
Line 148 ‘menaged’ ? clarify if this is technical term or use different word.
Line 178 (and other places) ‘relevant’ instead of saying relevant. explain why it is relevant e.g., commercially important/ commonly studied etc.
Line 181- 187: There are many methodological details missing here and this needs to be addressed to ensure replicability/reproducibility and for the interpretation of the results of behavioural assays. Specific examples are: what temperature was the water, what was the grain size, colour, depth of the sand provided, what were the dimensions of the shelter provided, and why was that particular object used as shelter (e.g. have previous studies shown fish use this form of shelter). Please see a recent guide for reporting these enrichment details Jones et al. (2021) [Note I am an author on this paper - please do not feel obliged to cite it - but please read it to address the reporting issues]
Line 184-185- careful of subjective anthropomorphisms ‘to feel safe’ - I would delete this part, do they feel safe there? How do we know? What does ‘feel’ mean for these fish?
Line 188 – why was three days used for acclimation?
Line 190 – when were the tests conducted, what was the actual measures scored, e.g. amount of time spent swimming at x speed, or x body length per y time, or movement vs no movement. When did trial start i.e. under what conditions? Did the fish need to be outside or inside the shelter, before or after feeding before a trial was started? If it was ad hoc trial start, how did you account for the initial position of a fish?
Line 192 – I assume tests were repeated daily 4 days per fish, but clearly state this.
Line 193 – how was exploration scored? How many trials were run, when? What was the toy. Lots of detail missing here.
Line 204 – why did you use fish from two populations. Just add a bit of explanation here even if it is something like ‘as per the design of the overarching project we collected fish from 2 locations.’
Line 219 – extra ‘I’
Line 231: what is a proposal here (see glossary suggestion)
Line 236 – how/ why where these 60 vids selected, you mention in the next sentence to cover the full range of video conditions – but add a bit more detail please. E.g. how many of each ‘condition’ and why did you chose that many.
Line 244-255. It is very important but expand a bit here on why it is important and connect to the number used in your study.
Line 255 - provide, drop the ‘s’.
Line 281 – why randomly select videos? Or rather why only randomly select videos, you recognise the potential impact that different video conditions (e.g. lighting etc) can have on image processing and detection etc. I would like to see a direct comparison of some selected videos with specific conditions (as per your selection on line 236)
Line 358 – word mixup, reword to something like : ‘significantly reduced the effort (time to score the videos)’
Line 359 - You cannot state this as you did not actually test whether ‘the accuracy and the precision of the analysis, is free from human errors.
Line 360 – 361 – I find this problematic: “we therefore have developed a novel algorithm that can help analysing the behaviour of a relevant species for aquaculture. Our Faster R-CNN”. From my reading, this is not your algorithm, and not novel. As you state just a few lines above, you adapted the Faster R-CNN.
Line 374 – Again, I would argue with the argument that this is a ‘new’ tool. It is more of an adaptation (for a very specific setup and species).
Line 385-373 – I really like this section of text that discussed to some of the interesting practical details and places your study in the context of similar work in this field.
Figure 2 ‘dorada’ maybe add for non-Spanish readers an additional clarifying description (sea bream or fish) to make it obvious to the casual reader what dorada means and connect it to the study species.
Figure 3: Add explanation of ‘DL’ and (especially) ‘supervised’ to the legend to make it stand on its own.
Figure 6: is hard to understand at first look. Consider adding some additional clarification to the legend e.g. something like ‘longer lines represent greater positioning error’.

Ref mentioned
Jones, N. A. R., Webster, M. M., & Salvanes, A. G. V. (2021). Physical enrichment research for captive fish: Time to focus on the DETAILS. Journal of Fish Biology, 99( 3), 704– 725. https://doi.org/10.1111/jfb.14773

Reviewer 2 ·

Basic reporting

The work entitled "Measuring laboratory fine-scale behavior in the gilthead seabream using deep learning" is very interesting. Despite the title, the work actually presents from a purely technical point of view the application of a deep learning algorithm to the recognition of a sea bream within an arena of behavior. It is understandable that the authors have chosen a simple behavior for this type of application, but the work is completely focused on the technical question of the algorithm. So in reality the work is not focused on the behavior of the sea bream, but simply on the algorithm performance. The Discussion is a very clear example that the behavior problematic in the sea bream is completely omitted. If the authors want to keep a title like this they should delve into these aspects and clearly highlight what the findings are regarding the behavior of the sea bream. Or perhaps the paper should be directed to a section of the journal specific to deep learning issues. This is a general comment for the rest below some minor comments.
85 the authors have to explain what mean semi-natural arena
88 the sentence is unclear
87-97 The hypothesis of the authors is that the light is the main problem for this type of study.
127-141 The aim of the study is unclear and vague
195-197 the procedure is not clearly explained
281 About the 14 videos sampled: are they regarding the activity, the exploration or both?
288-290 The sentence is unclear
301 presence or absence of the fish mean outside or inside the shelter?

Experimental design

no comment

Validity of the findings

no comment

Additional comments

no comment

Reviewer 3 ·

Basic reporting

- Although in general the article reads well, there are still quite a lot of textual mistakes in the text, including simple spelling mistakes (e.g. "pounds" instead of "ponds" , line 36), wrong grammar (e.g. " ..being usually its analysis time consuming and always subject to errors...", lines 65-66), bad English (e.g. "more information about what it is a fish than about what it is not a fish", line 387; "The time in seconds is on the x-axis and the probability of the detection of the fish is on the y one.", figure 3 caption), non-professional English (e.g. "the majority of them get only a little bit better", line 335) etc. The whole MS should be carefully reread and corrected by (someone with a similar level as) a native English speaker.
- The introduction and background provide relevant context
- The structure seems to be mostly conform PeerJ standards. I however found the other of the methods a bit unusual, starting with a description of the CNN, then discussing details about the animals and experiments, and then again how the CNN was used.
- The Figures are mostly relevant but: the captions for figure 4 and 6 should start with the x-axis not the y-axis and it should be clarified what the different videos are or why they are included; Figure 6 has much smaller axes so should be changed; figure 7 it is not clear enough in the axes the plot region is the tank borders, it is not clear where in the tank the refuge is, and it is not stated how much time each trajectory represents. It would also be very helpful to include some figure that shows the different augmentations or potentially difficult situations for the CNN.
- Raw data is supplied but I cannot find an easy explanation how the data is organised and what the data and filenames represent.

Experimental design

- I believe the research is within the scope of Peerj, although I question a bit if it can be labelled as "primary research" as it is mostly methodological
- It is clear what the authors intend to do with this article, their objective is meaningful, and it is clear what gap it fills in the literature, but I wouldn't call this a knowledge gap.
- The research seems to be performed using proper standards
- The methods are described in sufficient detail to replicate the experiments, but some more step-by-step (supplementary) descriptions would be needed to be able to also replicate the computational work performed that is central to the paper.

Validity of the findings

- The findings overall seem valid but actually no statistics have been done and only means and standard deviations (? this is actually not specified) provided and compared. I am not sure if a more quantitative and statistical approach is more appropriate here. The conclusions are based on the provided metrics and link back to the original research aims.

Additional comments

I agree that deep learning has much potential to help further automate behavioural observations under difficult and varying environmental conditions. But I think that the conditions used in the study, empty experimental tanks with a single refuge under standard lab conditions, are not the best to properly test the DL approach. As a matter of fact, I think the present videos generated with the used setup and conditions can be more quickly and easily analysed using tracking scripts/software compared to the used and proposed DL approach. Using computer vision it has become very easy to keep track of animals, especially individually, and even with shadows and changing backgrounds. I unfortunately do not see a clear way how the authors can address this point without collecting new data under much less standardized and uncontrolled conditions. To help clarify this point, the authors could revise their manuscript to better discuss the benefits of animal tracking versus using deep learning approaches and when each may be better and faster, including by providing details about the time it takes to analyse say 1h of video data using their deep-learning approach. The manuscript is also lacking a proper description and showdcase of the disturbances or potential difficulties for tracking using their setup. An actual comparison with simple tracking software that uses static or dynamic background substraction would be very valuable. Note also that the manuscript discusses the arenas as "semi-natural behavioural arenas" (line 35; 133) while criticising automated tracking studies to use "plain, white, homogeneous behavioural arenas" (line 83), while the tank used in the present study is actually also a rather blank tank with an artificial refugre (as is clear from Figure 1), so this wording needs to be changed.

The study is very focused on the deep learning method but actually does not provide much to the interested reader to help them on their way to also start using this approach. In my opinion the article could be much improved if some more background was provided about the different deep learning approaches and how researchers may go about using them, as well as considerations and potential simple-to-understand steps to employ a CNN. This is more of a suggestion as it is not about the potential readership of the paper, but also about its suitability to be published as it should be clear what time investments are really needed to set up, trial, test, augment, and validate neural networks.

The authors highlighted throughout the manuscript that they used "a novel Raspberry Pi high throughput recording system", but then actually do not provide much detail or any further attention to this point, such as why and how these computers were used. It therefore remains a bit unclear why this information is highlighted. Then in the last paragraph of the discussion the authors state they are working on incorporating deep learning on the Raspberry Pi to perform similar work. With this in mind, the manuscript could be improved by explaining the computational requirements of DL approaches and discussing the (future) potential for (real-time) analysis, like on single-board computers as the Raspberry Pi. Actually including some data on this in the paper, which the author seem to have (lines 455-458), and revising it accordingly (i.e. that this is possible using low-cost computers), would be valuable.

The authors set as their final aim "to serve as a tool for evaluating individual differences in fish behavioural types, also known as temperament or personalities, and to search for behavioural syndromes" (lines 128-130), but this is more the aim of their own research with this tool, not the aim with the present paper as no data is provided at all on this point. The authors should therefore revise their aims statement. I do however feel the article is rather methodology heavy and some more attention should be placed on to the actual behavioural data acquired and what types of (temporal) analyses that this enables.

---

## Round 0.2 · Minor Revisions

The reviewers have commented on your above paper. They indicated that it is not acceptable for publication in its present form. Your manuscript has been improved but there are still some underlying points from various reviewers that need to be considered. You should consider in the document how the CMM approach is more beneficial than more conventional approaches such as the use of CVs. The referees have also provided a few more considerations to be considered in the revised version.

Reviewer 1 ·

Basic reporting

I believe all basic reporting criteria are met. However, I have some final remarks concerning the ambiguity of the text.

I see all previous reviewers had some issue with the aims/ main message of the manuscript. It is tricky for the authors as this ms straddles two areas, and they have made a good attempt at re-focusing the manuscript, however there is still room for improvement. For example, the aims outlined on lines 123- 124. I think this important sentence could be far more precise. Currently the line says “..to automatically track a marine fish…” However, more specifically it was used to train and validate the ability to track several behavioural measures commonly used in for assays of individual differences (behavioural types /personality) e.g.: …including, 1) level of activity, 2) Shelter use and 3) Exploration of novel objects (neophobia).. . etc. I think this could be re-worked. Similarly, there are other places in the ms where the aims are mentioned, e.g. in the abstract, the methods and the first paragraph of the discussion each time with slightly different wording and each could be more focused on what the study actually aimed to and or achieved.

Relatedly, I think the title could be changed to be more reflective of the study. ‘Fine-scale measures’ is a bit too ambiguous. I might suggest linking it specifically to measures of inter-individual differences in behaviour. The abstract might also be amended to be more specific about the behavioural measures validated.

Also, please may the authors be more more consistent with their use of terminology. For example, behavioural types/ individual differences/personality/ behavioural variation and other terms were all used to refer to the same thing. I am not a fan of ‘personality’ and I like the authors use of behavioural types, but they should select one (or max two terms) and use them consistently in the ms. Otherwise it gets a bit harder for the reader to follow.

Experimental design

See above about clarity of the aims. I would like to say the authors made some significant changes to the methods and they are much clearer and more reproducable.

Validity of the findings

Good, however, again the conclusions would be more impactful if they are linked directly to more precisely described aims of the study.

Additional comments

Line 94 - Authors refer to fish here but until this point the ms has been forced more generally at animals - maybe change to animals here too. Or re-focus the intro to fish studies more explicitly earlier.

Line 97-99 add some minor details to the examples mentioned e.g. What species were involved?

Line 108-109. I don’t believe it is difficult to foresee at least some of the advancements this sort of technology can lead to. In fact, the authors clearly mention some of the benefits and changes we might expect and are already seeing at other places in the ms.…

Line 172 – Don’t forget to add in the actual sizes of the tanks ( the XXXXL)

Reviewer 2 ·

Basic reporting

The authors revised the manuscript, following all comments of the reviewers. Thus the manuscript is now ready to be published

Experimental design

Now the experimental design is clear

Validity of the findings

The manuscript in this version face much more aspects linked with the sea bream behavior not purely from a technical point of view.

Reviewer 3 ·

Basic reporting

As this is a review of a revision, see below.

Experimental design

As this is a review of a revision, see below.

Validity of the findings

As this is a review of a revision, see below.

Additional comments

The authors have improved their manuscript throughout and properly addressed many of the points raised by me and the other reviewers, but I still remain with my main concern that the manuscript does not make a clear enough case how the CMM approach is beneficial over more conventional approaches such as using CV. You have included a paragraph in the discussion stating your difficulties to use CV and in the introduction highlight some limitations, but this is not enough. Your statement that "computer vision tracking algorithms require plain and homogenous experimental conditions" (lines 174-175) is not correct as it is also possible to track objects in very heterogeneous environments and also in environments that change over time, such as by using a dynamic filter. It is also not too difficult to update the tracker when the environment has changed, such as when a novel object has been provided. Furthermore, given that the environment you used was also very static and controlled and homogenous I cannot see why the CMM would be much more beneficial. You do now state in your discussion you tried CV but failed to use it properly, but I would think that with a bit more effort CV could actually work.
You actually start your discussion by stating "Our neural network reduced significantly our time effort and increased the accuracy and the precision of the analysis, reducing human error" (lines 1358-1360). My key criticism about this is that you (still) do not provide any quantitative measures comparing the approach to an alternative approach such as using CV. So you can actually not make this statement. I asked in my original review already to provide more information about the time investment that is needed to set up, train, validate, and run the deep learning approach and highlighted its benefits to future readers but this was not properly addressed either as you still don't provide thorough details nor describe the costs and benefits of each approach. Currently the paper only shows the DL approach works relatively well but does not provide much consideration about why one would not use CV approaches. Related to this, I also commented to provide more background information for researchers to use and employ CNNs, including about the time investments needed, to which you replied that there are tons of detailed explanations and tutorials on the internet. This is not reallh satisfactory. I didn't mean that you should state in detail the exact steps how to use DL but to provide more context why to use it, what considerations to take into account, and what costs and benefits there are, for which I really recommend to better discuss some specific cases where conventional tracking or DL might be most benficicial. Let me clarify that I do see a lot of value in a DL approach and I think publishing a paper like this about it is valuable, I just think it is very limiting to only say it works without providing proper discussion of its costs and benefits.

Besides this major point, I just had a couple further comments:

You removed "semi-natural" and now use "enriched". This wording is not really appropriate either as you still used relatively blank tanks with just a shelter. That is not enrichment, as you state it on line 299.

For the Raspberry Pi you cite Gay, 2014. This is not the proper reference. You should cite the Raspberry Pi Foundation instead (e.g. Raspberry Pi Foundation, 2020, Raspberry Pi foundation annual review) or a relevant review that highlights its relevance and use.

You don't state from where the cameras filmed the fish. From in front or above?

Why did you for validation randomly sampled 11 from the activity and only 3 from the exploration trials?

You say that for validation you manually recorded the presence or absence of the fish at every second. Since the videos were one-hour each, do you mean you manually went through all 3600 frames per video?

Could you clarify how the trained dataset could be used in the future? Would you be able to run new tests using the same set up and species and then get an accuracy similar like the 92.79% you found? I think some more discussion of this (related to my main point) would help improve your paper.

---

## Round 0.3 · accepted · Accept

Dear Authors,

I am pleased to confirm that your paper has been accepted for publication in PeerJ.

Thank you for submitting your work to this journal.